# Derivative-Based Non-Target Identification of DNA-Reactive Impurities with Fragment Ion Filtering

**DOI:** 10.3390/molecules30193981

**Published:** 2025-10-04

**Authors:** Dongmei Zhang, Baojian Hang, Yiran Zhang, Pengfei You, Feng Shi, Liping Gong

**Affiliations:** 1Biological Products Research Division, Shandong Institute for Food and Drug Control, Xinluo Avenue No. 2749, Jinan 250101, China; zhangdm1000@163.com (D.Z.); swordhang@139.com (B.H.); ypfyy@126.com (P.Y.); 13953175123@139.com (F.S.); 2National Medical Products Administration Key Laboratory for Research and Evaluation of Generic Drugs, Jinan 250101, China; 3Industrial Technology Foundation Public Service Platform, Jinan 250101, China; 4Department of Chemical and Biomolecular Engineering, Johns Hopkins University, Baltimore, MD 21218, USA; yzhan774@jh.edu

**Keywords:** non-target screening, DNA-reactive impurities, derivation, fragment ions filtering

## Abstract

DNA direct reactive impurities (DDRIs) can react with nucleophilic sites of DNA, leading to mutations. The control strategies outlined in International Council for Harmonisation of Technical Requirements for Pharmaceuticals for Human Use (ICH) M7 are based on the known compound structure of DDRIs. Non-target screening of DDRIs in drugs is still challenging due to the diversity of the species and the poor stability. In this study, a derivatization reagent including a reactive group and report group was designed to screen DDRIs. Based on the electrophilic theory of chemical carcinogenesis, an amine reagent was used as a reactive group to interact with DDRIs. Two derivatization reagents, p-methoxyaniline and p-methoxybenzoyl-β-alaninamide, were employed, each containing different chromatographic modification groups to mitigate matrix effects. The derivatization products were analyzed by ultra-high-performance liquid chromatography coupled to high-resolution mass spectrometry (UPLC-HRMS). Non-target screening for DDRIs was achieved by product ions filtering of the report group.

## 1. Introduction

The DNA direct reactive impurities (DDRIs) are a kind of electrophilic reactant whose interaction with the electron-rich DNA is considered to lead to mutations and potentially cause cancer. Alkylation and acylation are two electrophilic substitution reactions. Alkylating agents include alkyl (C < 5) or benzyl ester of sulfuric, sulfonic, phosphoric, or phosphonic acid; N-methylol derivatives, sulfur or nitrogen mustard, epoxides and aziridines, aliphatic halogens, alkyl nitrite, R, β-unsaturated carbonyls, simple aldehyde, and quinones. Acylating agents include acyl halides, isocyanate and isothiocyanate groups, and β-lactones [1]. The evaluation strategy of DDRIs is as follows: (1) summarizing the actual and potential impurities most likely to arise during the synthesis, purification, and storage of the drug substance; (2) assessment of DDRIs based on structural alerts using the Quantitative Structure Activity Relationship (QSAR) software; and (3) in vitro and in vivo genotoxicity tests, including the bacterial reverse mutation assay (Ames test), micronucleus test, Pig-a assay, and animal bioassay [2]. The current evaluation strategy is based on the known compound structure of DDRIs.

Screening for unknown DDRIs presents significant challenges in pharmaceutical development, primarily due to the structural diversity of these species and the inherent instability of their characteristic functional groups (e.g., alkylating agents, epoxides, or Michael acceptors). These impurities can originate in two ways: first, residual byproducts from chemical synthesis (e.g., incomplete purification of intermediates); second, through subsequent degradation pathways (hydrolysis, oxidation, or photolysis) triggered by reactive process components such as genotoxic reagents, organic solvents, or metal catalysts. A notable example is the 2007 global recall of the HIV drug Viracept^®^ (nelfinavir mesylate). Unexpected ethanol contamination in methanesulfonic acid (a starting material) led to in situ formation of high levels of ethyl methanesulfonate (EMS), a known mutagenic alkylating agent. This incident underscored the critical need for rigorous control of potential DDRIs throughout the drug manufacturing process [3].

Advanced analytical strategies are essential, as even trace-level (ppm/ppb) DDRIs can induce DNA adducts or chromosomal aberrations. Derivatization can be used to convert them into chemical forms more compatible with the chromatographic environment [4,5]. Modern hyphenated techniques (e.g., LC-HRMS, GC-MS/MS) coupled with derivatization methods [6,7] or predictive in silico toxicology tools (e.g., DEREK Nexus) allow for sensitive detection and structural elucidation of reactive impurities [8]. Furthermore, Quality by Design (QbD) principles and forced degradation studies should be implemented proactively to identify impurity formation risks during process development. Recently, the Orbitrap analytical platform has been used to screen compounds with similar structures [9]. However, matrix effects, where co-eluting components suppress ion signals, pose significant challenges in ultra-high-performance liquid chromatography and high-resolution mass spectrometry (UPLC-HRMS) analyses. These effects compromise the accuracy, sensitivity, and reproducibility of results [10,11]. Traditional sample preparation methods (e.g., extraction, clean-up) are less effective in mitigating matrix effects in non-target screening. These techniques are typically optimized for specific analytes and may inadvertently remove or alter unknown compounds of interest in untargeted analyses.

In this study, a targeted derivatization strategy was employed to enhance the detection and characterization of DNA direct reactive impurities (DDRIs) using amine-based reagents as derivatization agents [12,13,14]. In terms of derivatization reagent universality, van Wijk et al.’s reagents (4-dimethylaminopyridine, 1-(4-pyridyl)piperidine-4-carboxylic acid n-butyl ester) are only specific to alkyl halide-type alkylating agents and fail to react with acylating agents or epoxides due to single reaction sites/mechanisms [12]. In contrast, our unified amino reactive moiety enables reactions with alkylating (nucleophilic substitution), acylating (amidation), and epoxide (ring-opening addition) agents, achieving comprehensive DDRIs capture. Amino-reactive moiety enables reactions with alkylating (nucleophilic substitution), acylating (amidation), and epoxide (ring-opening addition) agents, achieving comprehensive DDRIs capture. Specifically, two structurally distinct reagents—p-methoxyaniline and p-methoxybenzoyl-β-alaninamide—were selected to systematically modify DDRIs through nucleophilic reactions (e.g., with electrophilic functionalities such as alkyl halides or epoxides). These reagents were designed with complementary chromatographic modification groups (hydrophilic aromatic amines vs. hydrophobic benzamide moieties) to address analytical challenges.

## 2. Results and Discussions

### 2.1. Principle of the Screening Strategy

Non-targeted screening and quantification of DDRIs in pharmaceuticals constitute a critical challenge in the biopharmaceutical sector [15,16,17]. DDRIs pose a significant risk as they have the potential to cause mutations and, consequently, cancer due to their interaction with electron-rich sites in DNA [18,19,20]. Following Miller’s electrophilic theory of chemical carcinogenesis, nitrogen of pyrimidine and purine in DNA are the main nucleophilic centers in DNA that potentially could interact with electrophilic species [1]. However, the product of this interaction arises from a wide and complex range of sources, including not only the structures of known compounds but also various unknown impurities. Traditional strategies for evaluating DDRIs rely heavily on known compound structures, but these methods are limited in detecting unknown impurities.

In this experimental design (Figure 1), amine groups were strategically employed as reactive groups in the derivatization reagents. Notably, compounds with identical reporter groups in their derivatization products exhibit conserved fragmentation patterns during collision-induced dissociation. This characteristic fragmentation generates a reproducible series of reporter-specific product ions, which act as diagnostic signatures. These signature ions enable reliable identification and selective detection of derivatized compounds through their distinctive mass spectral fingerprints.

### 2.2. Synthesis of Modular Derivatization Reagents

Matrix effects and ion suppression of co-eluting components are key challenges in LC-MS that can impact the accuracy, sensitivity, and reproducibility [21,22]. Modular derivatization reagents include reactive groups, reporter groups, and chromatographic modified groups. The strategic selection of orthogonal chromatographic modifiers enables spatial–temporal separation of analytes from matrix interferences via differential retention mechanisms. Orthogonal protection and modular strategies are widely employed in peptide synthesis [23,24]. By analogy, we engineered dual derivatization reagents with chemically orthogonal retention profiles to decouple target analytes from co-eluting contaminants.

To mitigate matrix effects, derivatization reagents with chromatographic modification groups were synthesized by coupling different functional modules using p-methoxyaniline as the starting material (Figure 2A). The structure of p-methoxybenzoyl-β-alaninamide was confirmed by nuclear magnetic resonance spectroscopy (Figure 2B).

### 2.3. Separation of Derivatization Product

Both p-methoxyaniline and p-methoxybenzoyl-β-alaninamide were used as derivatization reagents. The methylated derivatives of these reagents were prepared according to the method described in Section 3.3 using 100 μg/mL methyl methanesulfonate. Derivatization products of different DDRIs were subsequently analyzed in the UPLC-HRMS experiment. A reversed-phase stationary, such as C18, was used to separate derivatization reagents from derivatization products. As shown in Figure 3, chromatographic analysis demonstrated clear separation between the derivatization reagents and their methylated derivatives. Native p-methoxyaniline eluted at 5.36 min, while its methylated counterpart (methyl-p-methoxyaniline) exhibited a significantly prolonged retention time of 10.44 min, corresponding to a ΔRT of +5.08 min. Similarly, p-methoxybenzoyl-β-alaninamide and its methylated derivative (methyl-p-methoxybenzoyl-β-alaninamide) showed retention times of 12.60 min and 13.86 min, respectively, with a ΔRT of +1.26 min. The observed retention time shifts (ΔRT = +5.08 min for p-methoxyaniline and +1.26 min for p-methoxybenzoyl-β-alaninamide) are consistent with the hydrophobic retention mechanism of C18 reversed phase chromatography, where the increased hydrophobicity of methylated derivatives enhances their interaction with the stationary phase.

This dual-reagent strategy not only ensured clear separation of derivatized products in UPLC-HRMS (retention time shift > 2 min) but also minimized co-elution interference through complementary retention behaviors. The orthogonal chromatographic selectivity of these reagents effectively differentiated target analytes from matrix components, enhancing sensitivity by 3–5-fold while reducing ion suppression effects in complex pharmaceutical matrices. Meanwhile, this mechanism ensures effective separation between the derivatization reagents, and most derivatization products can be achieved by reversed phase stationary such as C18. Furthermore, distinct retention profiles of these methylated derivatives can partially mitigate matrix effects by minimizing co-elution with interfering compounds in complex samples. Notably, the active pharmaceutical ingredient (API) has an extremely high concentration and must be diverted to waste liquid in mass spectrometry analysis. The distinct retention time windows of the two derivatization systems enable orthogonal separation capabilities to partially mitigate the interference of API.

### 2.4. Optimization of Derivatization Reagent Concentration

To optimize the derivatization reagent dosage, reaction experiments between EMS and the derivatization reagent were carried out to explore the relationship between the amount of derivatization reagent and the reaction performance of reactive impurities. In a 200 μL reaction system, the EMS final concentration was adjusted to 1 ng/mL, while the final concentrations of the derivatization reagent were set to 20 mg/mL, 10 mg/mL, 1 mg/mL, 0.01 mg/mL, and 0.01 mg/mL. After a 30-min reaction, the mass spectrometric responses of the derivatization products were measured. Results showed that at reagent concentrations of 20 mg/mL or 10 mg/mL, reaction efficiency approached 100%; the reaction efficiency was close to 100%, with almost no difference between them. Considering both reagent usage and the reaction effect, 10 mg/mL was finally selected as the concentration of the derivatization reagent in the derivatization reaction (Figure 4).

### 2.5. Solvent Selection in Derivatization System

Solvents significantly impact reaction efficiency [25,26]. To investigate these effects, DMSO, DMF, ACN, THF, and physiological environment (PBS (phosphate-buffered saline), pH 7.4, 0.01 M) were tested for derivatization reactions. Optimal conditions were different for different DDRIs: aprotic polar solvent is preferred for alkyl ester of sulphonic, alkyl halides, and acyl halides; epoxy reacts with amines only in a physiological environment; acyl chloride can react instantly with water, so that derivatization with acyl chloride in a physiological environment is impossible. In the derivatization of aldehyde, first a Schiff base is formed, and sodium cyanoborohydride (NaBH_3_CN) is needed to reduce the Schiff base to a single and more stable secondary amine product.

Reactivity of derivatization reaction with amine is generally associated with nucleophilic reaction with DNA [27,28,29]. Impurity genotoxicity was related to the reactivity of the impurity with electron rich groups in DNA. Chemical reactivity has often been used as a tool to study carcinogenicity [30,31,32]. Although physiological environment is not the optimal reaction condition for all impurities, physiological environment is the reaction condition of DDRIs with DNA. If a substance (e.g., acyl halides) is too reactive, it will actually react with water rather than nucleophilic centers in DNA [32]. Physiological environment can be used as the reaction condition to assess the genotoxicity of DDRIs in APIs by the drug administrative agency. Different derivation solvents can be used for the determination of DDRIs by drug manufacturers (Table 1).

### 2.6. Fragmentation Patterns Analysis for Derivatization Product

Characteristic product ions of different derivatization products with different DDRIs are shown in Table 2, where *m*/*z* 123.0680 was detected in MS2 spectra of alkyl p-methoxyaniline; *m*/*z* 178.0865 was detected in MS2 spectra of acyl p-methoxyaniline; *m*/*z* 124.0757 and *m*/*z* 166.0863 were detected in MS2 spectra of alkyl p-methoxybenzoyl-β-alaninamide; and *m*/*z* 178.0865 was detected in MS2 spectra of acyl p-methoxybenzoyl-β-alaninamide. These main product ions mentioned above were used to reconstruct the structure of the derivatization reagent for fishing derivatization products using Compound Discover 3.0. In full MS/dd-MS2 (Top N) mode, the Top N ions in MS1 were selected for analysis in MS2. The parent ion of the characteristic fragmentation was easily identified. The dynamic exclusion list included the MS1 ion, which was found in the sample, and the derivatization reagent can be established to reduce matrix effects.

### 2.7. Method Validation

We systematically evaluated the non-target screening limits of ethyl methanesulfonate (EMS) in nelfinavir mesylate using an Orbitrap HRMS platform operating in full MS/dd-MS2 (Top N) mode. For this evaluation, EMS was spiked into the nelfinavir mesylate matrix at three concentration levels: low (1 μg/g), medium (10 μg/g), and high (50 μg/g). The method demonstrated distinct detection sensitivities for two derivatization reagents: p-methoxyaniline achieved a 50 ppm detection limit, whereas p-methoxybenzoyl-β-alaninamide exhibited enhanced sensitivity with a 10 ppm detection limit. Notably, a key limitation of trapping-based mass spectrometers is the limited charge capacity of the ion trap, which excludes many low abundance ions from MS1-level analysis. Chromatographic separation resolved ethylated derivatives of the two reagents at 13.80 min and 15.85 min, respectively, with baseline resolution (>2.0). Dynamic ion injection profiles (Figure 5) revealed contrasting behaviors. At 13.8 min, a high-intensity signal (2.79 × 10^7^) required only 6.6 ms of injection time. For the ion *m*/*z* 152.1065, which corresponded to the ethylation product of p-methoxyaniline, the signal-to-noise ratio (S/N) was 2.3 under the condition of 10 ppm EMS. In contrast, the eluent at 15.85 min had a reduced intensity (6.29 × 10^6^), even with a prolonged injection time of 39.0 ms, the S/N for the ion *m*/*z* 223.1439, corresponding to the ethylation product of p-methoxybenzoyl-β-alaninamide, reached 30.5 at 10 ppm EMS. Extending injection time can increase the injection volume of trace substances, improving the detection efficiency. The fivefold sensitivity difference between the reagents may originate from matrix effects: matrix components co-eluting with p-methoxyaniline derivatives preferentially occupy limited ion trap capacity, while the greater hydrophobicity of p-methoxybenzoyl-β-alaninamide shifted elution to a cleaner chromatographic window. Strategic selection of derivatization reagents with tailored retention properties effectively minimizes co-elution with interfering species, providing a viable solution to mitigate charge capacity constraints in trapping-based mass spectrometry [33].

### 2.8. Sample Detection

During the risk assessment of drug samples, a batch of fasudil samples was found to contain methylated and ethylated substances. Analysis revealed a potential source: fasudil hydrochloride is synthesized using 5-isoquinolinesulfonic acid as the raw material, with N,N-dimethylformamide as the catalyst. The synthesis process involves reaction with thionyl chloride, followed by alkalization, substitution reaction with homopiperazine, vacuum concentration, and final purification. In this reaction sequence, a certain residual amount of 5-isoquinolinesulfonyl chloride hydrochloride generated cannot fully react with homopiperazine and remains in the reaction mixture. During the recrystallization and purification process, the residual 5-isoquinolinesulfonyl chloride hydrochloride can react with methanol or ethanol in the recrystallization solvent to form methyl 5-isoquinolinesulfonate or ethyl 5-isoquinolinesulfonate. Subsequently, a UPLC-MS/MS-based method was established for the determination of ethyl 5-isoquinolinesulfonate content.

### 2.9. Quantitative Detection of Alkylated 5-Isoquinolinesulfonate by UPLC-MS/MS

Based on the above research, the UPLC-MS/MS method was developed to determine two alkyl sulfonate genotoxic impurities in fasudil hydrochloride. Separation was performed on a Waters ACQUITY UPLC BEH C18 column (50 mm × 2.0 mm, 1.7 μm) using an isocratic mobile phase of 0.1% formic acid in water and 0.1% formic acid in methanol at a flow rate of 0.3 mL/min. The column temperature was set at 40 °C with a 1 μL injection volume. Methyl 5-isoquinoline sulfonate and ethyl 5-isoquinoline sulfonate were analyzed using positive electrospray ionization (ESI+) and multiple reaction monitoring (MRM). Both compounds exhibited good linear relationships within the range of 2.05–205.32 ng/mL and 1.90–190.24 ng/mL (r = 0.9997). Detection limits were 0.51 ng/mL and 0.48 ng/mL, with quantification limits of 2.05 ng/mL and 1.90 ng/mL for the methyl and ethyl derivatives, respectively. Spike recoveries (*n* = 6) for both compounds ranged from 102.84% to 107.11%. This UPLC-MS/MS method is sensitive, accurate, and reliable for the quantification of methyl 5-isoquinoline sulfonate and ethyl 5-isoquinoline sulfonate in fasudil hydrochloride, providing a robust approach for monitoring these genotoxic impurities. This study designed derivatization reagents comprising reactive groups and reporter groups for the screening of DDRIs. However, several considerations remain for future research and development. First, the reaction efficiencies of different DDRIs may vary depending on the solvent used in the derivatization system. Therefore, pharmaceutical manufacturers and R&D institutions should conduct impurity screening under multiple reaction conditions to ensure the detection of a wider range of DDRIs. Second, while the physiological environment (PBS) may not be the optimal reaction condition for all impurities, it is important to consider this condition as it mimics the in vivo environment where genotoxic impurities interact with DNA. Drug regulatory agencies should prioritize the use of PBS as the reaction solvent to better assess the genotoxicity of DDRIs in APIs. Furthermore, continuous efforts are directed towards synthesizing more modular derivatization reagents. Incorporating mass spectrometry signal-enhancing groups can enhance the mass spectrometry response. Additionally, the inclusion of diverse chromatographic modification groups will be beneficial in overcoming matrix effects, thereby enhancing the discovery of unknown impurities.

## 3. Materials and Methods

### 3.1. Chemicals and Reagents

p-methoxyaniline, methyl methanesulfonate, ethyl methanesulfonate (EMS), butyl methanesulfonate, pivaloyl chloride, 2-thiophencacetyl chloride, 1,2-epoxy-2-methylpropane, (R)-2-oxiranylanisole, aldehyde, phenyl aldehyde, and piperidine were purchased from J&K Scientific Ltd. (Beijing, China). Acetonitrile (ACN), formic acid (FA), dimethyl sulfoxide (DMSO), dimethylformamide (DMF), tetrahydrofuran (THF), Fmoc-β-Ala-OH, N,N′-Dicyclohexylcarbodiimide(DCC), N-Hydroxysuccinimide(HOSU), and Dichloromethane(DCM) were purchased from Sigma-Aldrich (St. Louis, MO, USA).

### 3.2. Synthesis Route of p-Methoxybenzoyl-β-alaninamide

5 mmol Fmoc-β-Ala-OH was dissolved in 20 mL dichloromethane. 5.5 mmol DCC was added with ice-bath cooling and stirring. Separately, 5.5 mmol HOSU in 10 mL DCM was slowly added dropwise. After 30 min on ice and 3 h at room temperature, insoluble matter was filtered out. The filtrate was rotary evaporated to obtain Fmoc-β-Ala-OSU. An amount of 1 mmol of it was dissolved in 10 mL THF with 2 mmol triethylamine on ice, then 1 mmol p-methoxyaniline was added. After 4 h at room temperature, followed by the addition of 20% piperidine to remove the Fmoc protecting group, the solvent was removed. The crude product was purified by reverse-phase HPLC.

### 3.3. Derivatization Reaction

In total, 100 μL of derivatization reagents (10 mg/mL) with buffer solution was added into a 1.5 mL tube. Subsequently, 100 μL of the sample solution was added. DMSO was used as an alternative solvent to dissolve poorly soluble samples in water. The mixture was incubated at 37 °C for 16 h with shaking at 150 rpm. Then 50 μL of the solution was used for analysis by UPLC-HRMS.

### 3.4. UPLC-HRMS

Analysis of samples was performed on UPLC-HRMS consisting of UltiMate 3000 HPLC and an Orbitrap Q-Exactive plus with an ESI (Thermo Fisher Scientific, Waltham, MA, USA). The high-performance liquid chromatography (HPLC) separation was performed on a column (Acclaim™ 120 C18 250 mm × 2.1 mm., 5 μm, Thermo Fisher Scientific, Waltham, MA, USA) at a flow rate of 0.3 mL/min at 40 °C. Formic acid in water (0.1%, *v*/*v*, solvent A) and formic acid in acetonitrile (0.1%, *v*/*v*, solvent B) were employed as mobile phases. Gradients of 0–4 min 2% B, 4–105 min 2% to 95% B, 105.1–110 min 95% to 2% B, and 110.1–120 min 2% B were used. Samples were detected under positive ion mode. Full MS followed by ddMS2 mode was used; the full MS scan range was set from *m*/*z* derivatization reagent +10 to 1000, Top N is 10, max injection time is 50 ms, and the dynamic exclusion list includes MS1 ion found in sample and derivatization reagent.

### 3.5. Data Analysis

The Compound Discover 3.0 (Thermo Fisher Scientific, Waltham, MA, USA) node “Compound Class Scoring” was applied to screen derivatization product by characteristic product ions.

### 3.6. UPLC-MS/MS

Quantitative detection of alkylated 5-isoquinolinesulfonate was carried out using an AB SCIEX Triple Quad 6500+ ultra-performance liquid chromatography–triple quadrupole mass spectrometry system (AB SCIEX, Framingham, MA, USA). The chromatographic column was an ACQUITY UPLC BEH C18 column (50 mm × 2.1 mm, 1.7 μm, Waters, Milford, MA, USA). The mobile phase consisted of a 0.1% formic acid aqueous solution—0.1% formic acid methanol solution (60:40). The flow rate was 0.3 mL/min, the column temperature was set at 40 °C, and the injection volume was 1 μL. The separation was achieved by isocratic elution. The ionization source was an electrospray ionization source (ESI) operating in the positive ion scan mode, with multiple reaction monitoring (MRM). The ion source temperature was 550 °C. The specific mass spectrometry parameters for 5-isoquinoline sulfonic acid methyl ester and 5-isoquinoline sulfonic acid ethyl ester are as follows: for 5-isoquinoline sulfonic acid methyl ester, parent ion *m*/*z* 224.100, daughter ions *m*/*z* 129.0 (cone voltage 50 V, collision voltage 40 eV), 102.1 (cone voltage 50 V, collision voltage 63 eV); for 5-isoquinoline sulfonic acid ethyl ester, parent ion *m*/*z* 238.000, daughter ions *m*/*z* 210.0 (cone voltage 40 V, collision voltage 35 eV), 129.0 (cone voltage 40 V, collision voltage 45 eV).

## 4. Conclusions

It is difficult to predict unanticipated DNA direct reactive impurities during the manufacturing process based on current industry practice. Common industry practices may not always be consistent with the CGMP requirements, and the manufacturers are responsible for the quality of their drugs. In this paper, we described a simple derivatization–HPLC–HRMS method for non-target screening of direct DNA-reactive impurities using an amine reagent as the reactive moiety for reaction with DDRIs using derivatization reagents containing different chromatographic modifying groups—p-methoxyaniline and p-methoxybenzoyl-β-alaninamide. Importantly, the impurity types screened by this method cover most DDRI categories specified in the ICH M7 guidelines, effectively addressing the limitation that ICH M7 currently lacks approaches for screening unknown impurities. The identification of unknown impurities and analysis of their sources can provide critical insights for optimizing drug manufacturing design—similar to how the genotoxic impurity incident of nelfinavir mesylate revealed that sulfonate drugs are prone to generating sulfonate esters, prompting enhanced control over alcohol solvent residues in sulfonate drug production. This study’s analytical capability thus helps reduce the risk of unknown impurity formation, laying a foundation for integrating the method into quality-by-design practices. To validate the utility of our approach, methylated and ethylated substances were detected in a batch of fasudil samples. By analyzing the synthesis process, we identified these substances as methyl 5-isoquinolinesulfonate and ethyl 5-isoquinolinesulfonate. Subsequently, a quantitative method for determining the content of these impurities in fasudil hydrochloride was established using UPLC-MS/MS. This method provides an extra dimension to the current industry practice to evaluate unanticipated DNA direct reactive impurities.

## Figures and Tables

**Figure 1 molecules-30-03981-f001:**
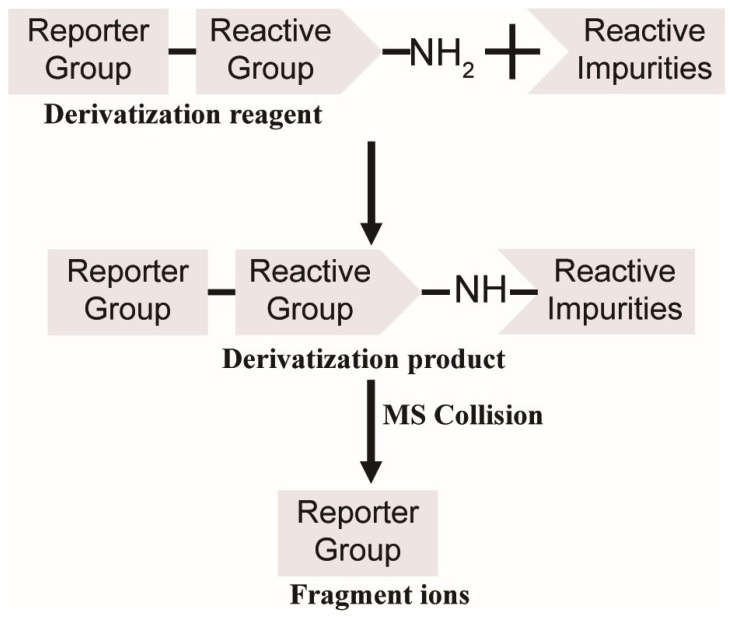
Strategy of non-target screening for direct DNA-reactive impurities using derivation and fragment ion filtering.

**Figure 2 molecules-30-03981-f002:**
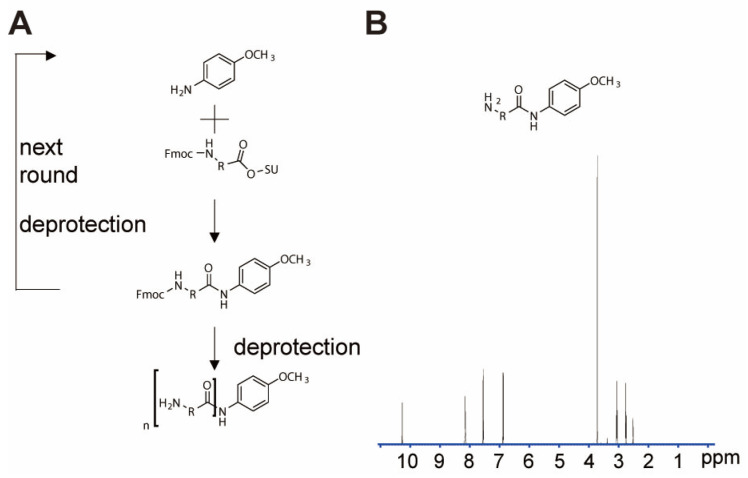
Synthesis of modular derivatization reagents: (**A**) synthesis route of derivatization reagent; (**B**) NMR identification results of p-methoxybenzoyl-β-alaninamide.

**Figure 3 molecules-30-03981-f003:**
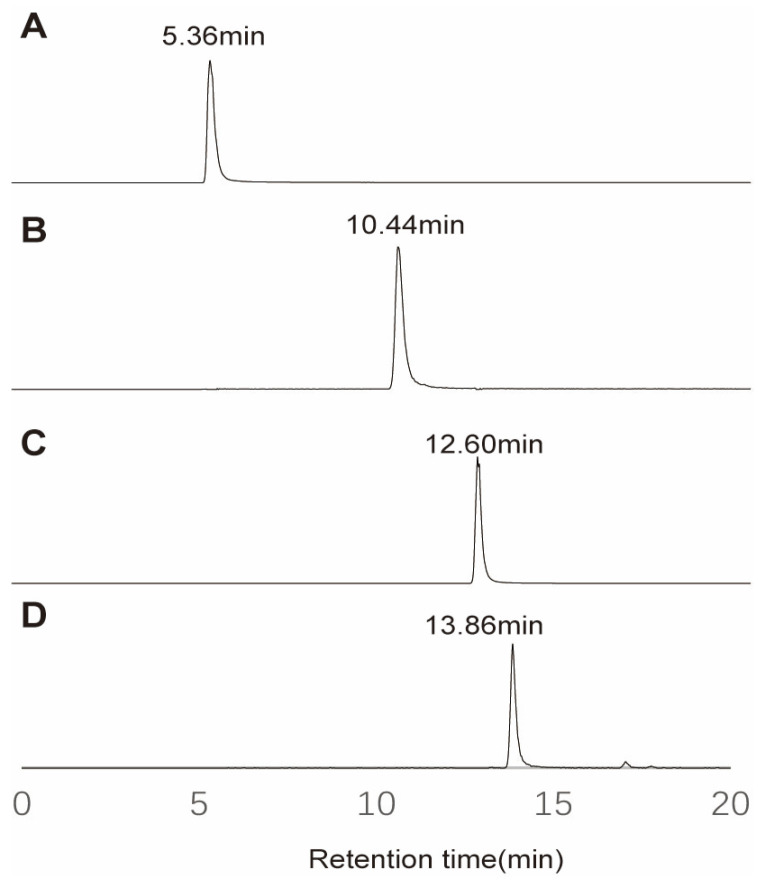
Retention time of derivatization reagent and derivatization product. (**A**) p-methoxyaniline, (**B**) methylated p-methoxyaniline, (**C**) p-methoxybenzoy-B-alaninamide, (**D**) methylated p-methoxybenzoy-B-alaninamide.

**Figure 4 molecules-30-03981-f004:**
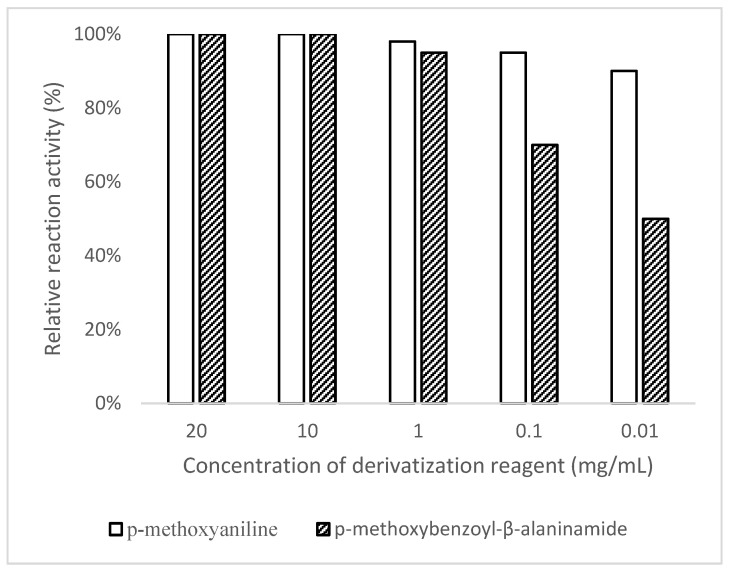
Dosage study of derivatization reagent in reaction.

**Figure 5 molecules-30-03981-f005:**
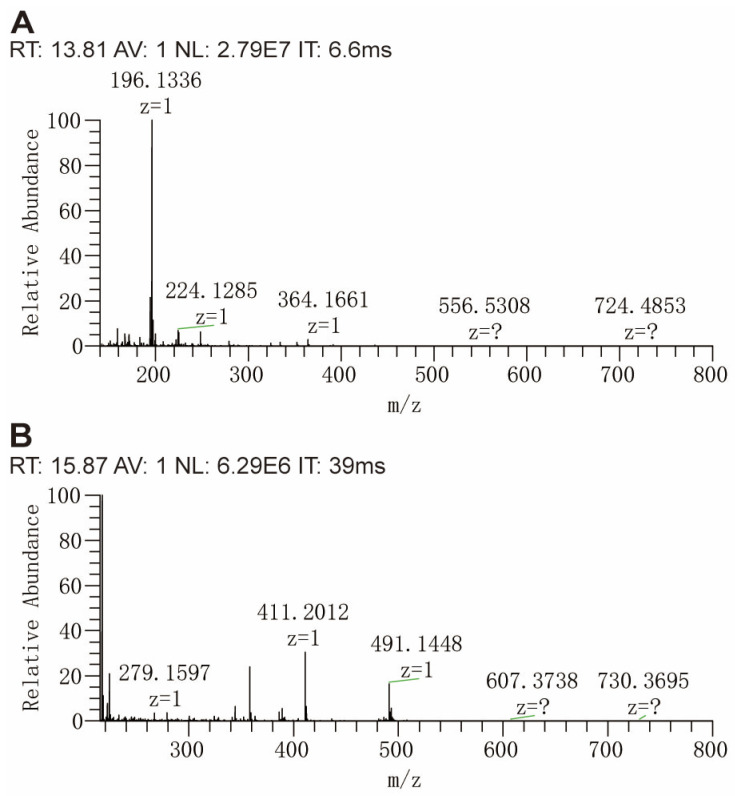
Full MS scan of nelfinavir mesylate at 13.8 min (**A**) and 15.85 min (**B**). ‘?’ indicates that the charge number z is uncertain.

**Table 1 molecules-30-03981-t001:** Solvent effects for the derivatization.

DDRIs	Relative Reaction Activity (%) ^a^				
	ACN	DMSO	DMF	THF	PBS (pH 7.4, 0.1 M)
Ethyl methanesulfonate	100 ± 3.2	99 ± 4.1	97 ± 4.1	95 ± 3.6	70 ± 2.7
1,2-Epoxy-2-methylpropane	ND	ND	ND	ND	100 ± 3.1
Pivaloyl chloride	100 ± 2.2	98 ± 3.8	99 ± 3.3	97 ± 1.4	ND

^a^ The highest reaction reactivity in different solvents was set at 100%. Values are means ± standard error of three replicates. ND, not detectable.

**Table 2 molecules-30-03981-t002:** Retention time and characteristic product ions of different derivatization products.

Reagents	Derivatization Product of p-Methoxyaniline		Derivatization Product of p-Methoxybenzoyl-β-alaninamide	
	RT (min)	Fragment Ions (*m*/*z*)	RT (min)	Fragment Ions (*m*/*z*)
Derivatization reagent	5.36	/	12.60	/
Methyl methanesulfonate	10.44	123.0680 124.0714	13.86	124.0757 166.0863
Ethyl methanesulfonate	13.80	123.0680	15.85	124.0757 166.0863
Butyl methanesulfonate	23.27	123.0680	20.07	124.0757 166.0863 148.0757 195.1128
Pivaloyl chloride	40.39	123.0680	36.11	178.0865
2-Thiophencacetyl chloride	43.16	124.0758	37.16	178.0865
1,2-Epoxy-2-methylpropane	8.25	123.0680	7.31	166.0863 178.0865
(R)-2-Oxiranylanisole	17.13	123.0680	13.57	124.0757 166.0863 180.1019
Aldehyde	4.59	123.0680	5.64	124.0757 166.0863

## Data Availability

All data generated or analyzed during this study are available from corresponding authors on reasonable request.

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
