# Peer review of "Derivative-Based Non-Target Identification of DNA-Reactive Impurities with Fragment Ion Filtering"

_molecules, 2025, doi:10.3390/molecules30193981_

Round 1

Reviewer 1 Report

Comments and Suggestions for Authors

Zhang et al. present a study entitled “Derivative-based non-target identification of DNA-reactive impurities with fragment ion filtering”. This work describes a derivatization strategy combined with UPLC-HRMS and fragment ion filtering to improve the non-target detection of DNA-reactive impurities (DDRIs). The authors used two complementary derivatization reagents (p-methoxyaniline, p-methoxybenzoyl-β-alaninamide) to reduce matrix effects and increase detection sensitivity. They tested the method with model compounds such as ethyl methanesulfonate and applied it to real pharmaceutical samples (fasudil), where methylated and ethylated isoquinolinesulfonates were found. The study highlights an important regulatory issue in pharmaceutical development, the detection of unexpected genotoxic impurities, and provides a useful proof of concept that extends impurity control beyond structure-based approaches.

Major Points

  1. Innovation and Contribution

The idea of using derivatization with common reporter groups to improve fragment ion filtering is an interesting and useful part of this paper. However, the novelty is not as strong as the authors suggest because similar derivatization-MS methods have already been published. The paper would be stronger if it directly compared this method with existing approaches, such as direct HRMS or other derivatization reagents.

  1. Validation Scope

The method was tested only with a small number of DDRIs and one drug example (fasudil). To show that the method is more widely applicable, the authors should test it with a broader range of impurities and more real-world pharmaceutical samples.

  1. Quantitative Rigor

The authors claim sensitivity improvements (3-5 fold), but they do not provide enough statistical support, such as confidence intervals, reproducibility across repeated experiments, or variation between different test days. Without this, the reported detection limits are less convincing.

  1. Discussion Depth

The paper mentions regulatory importance but does not explain clearly how this method could be connected to ICH M7 guidelines or integrated into quality-by-design practices. A deeper discussion of these practical applications would make the paper more impactful.

Minor Points

  1. Some figures like 3 and 5 are too dense and need clearer labels and shorter legends
  2. Terms such as DDRI and GTI are used inconsistently so terminology should be unified
  3. The paper uses many older references and should add more recent studies on non targeted analysis and exposomics
  4. The text is wordy in places and should be more concise for easier reading

Reviewer 2 Report

Comments and Suggestions for Authors

The submitted study reports the development of two derivatization reagents for the detection of DNA direct reactive impurities (DDRIs), which may cause alkylation or acylation of DNA present in pharmaceutical products. The dual-reagent design appears to be effective in mitigating matrix effects in mass spectrometric analysis. However, several concerns outlined below require substantial revision to improve the clarity and reproducibility of the manuscript.

1. In this study, two derivatization reagents, p-methoxyaniline and p-methoxybenzoyl-β-alaninamide, were employed. These reagents contain an aromatic amine and an aliphatic amine, respectively, and are therefore expected to exhibit different reactivities toward electrophilic DDRIs. If the use of two reagents with distinct reactivity profiles was intentional, please clarify the rationale behind this choice.

2. The sections titled ' 2.8 Sample Detection' and ' 2.9 Quantitative detection of alkylated 5-isoquinolinesulfonate by UPLC-MS/MS' appear to be unrelated to the main focus of the submitted study, which centers on the development and evaluation of derivatization reagents. Since these sections do not involve derivatization, please reconsider whether their inclusion is essential to the scope and coherence of the manuscript.

3. Section 2.3 of the manuscript describes the detection of methylated derivatives of the derivatization reagents. However, the experimental conditions under which the methylation reactions were performed (e.g., reagents, concentrations, reaction time, temperature) are not clearly stated. Please provide detailed information regarding the methylation procedure to ensure reproducibility and clarity.

4. Please carefully review the results presented in Table 2. For example, although pivaloyl chloride is an acylating reagent, the derivative of p-methoxyaniline is listed with an m/z of 123.0680, which corresponds to the unmodified or alkylated form, rather than the expected acylated form with m/z 178.0865. To improve clarity regarding the resulting product ions, it would also be helpful to include product ion spectra.

5. The submitted study evaluates the non-target screening limits of ethyl methanesulfonate (EMS) in nelfinavir mesylate. However, it is unclear whether EMS was used as a standard compound spiked into the matrix, or detected as an inherent impurity in the sample. In particular, the manuscript does not clearly describe how EMS was introduced into the derivatization reagents or under what conditions the reaction was carried out. To ensure clarity and reproducibility, please provide detailed information regarding the EMS derivatization procedure.

Reviewer 3 Report

Comments and Suggestions for Authors

In this study, the authors designed and synthesized two derivatization reagents, p-methoxyaniline and p-methoxybenzoyl-β-alaninamide, as reactive groups to interact with DNA direct reactive impurities. The derivatization products were analyzed by ultrahigh-performance liquid chromatography coupled to high-resolution mass spectrometry (UPLC-HRMS). 

The overall work presented is promising and should be of interest to the readership of the journal Molecules. Here are a few comments:

From a synthetic point of view, since the authors describe the synthesis of compounds to be used as derivatizing agents, detailed NMR spectra should be provided in supporting information for each intermediate or at least for the final compounds, as well as NMR interpretation for 1H-NMR and 13C-NMR. The synthesized compounds should be fully characterized and yields for each step in the synthetic route should be provided.

Figure 2 should be provided in higher quality. Bonds in chemical structures appear to have strange angles and should be corrected.

Comments on the Quality of English Language

Should be checked for minor errors

Round 2

Reviewer 1 Report

Comments and Suggestions for Authors

The revision is improved with updated references, clearer explanation of novelty, better regulatory context, and clearer figures and terminology.
However, the validation remains limited to a few DDRIs and one drug, and more replicates or statistical data would strengthen the conclusions. The English is understandable but could be polished for clarity.
Overall, this is a useful contribution and I recommend acceptance after minor revisions.

Reviewer 2 Report

Comments and Suggestions for Authors

The authors have provided thoughtful responses to the previous comments and have appropriately revised the manuscript. Therefore, I consider the revised manuscript acceptable for publication in its present form.

Author Response

Thank you very much for taking the time to review this manuscript.